# Exact Convex Reformulations of Linear Neural Networks via Completely Positive Lifting

## Abstract

We show that the training problem of a deep linear neural network under the squared loss admits an exact convex reformulation in a lifted space over a generalized completely positive cone. The reformulation has the same optimal value as the original nonconvex problem and is linear in the lifted variables, with all nonconvexity encoded in the cone constraint. Its ambient lifted dimension depends only on the input and output dimensions, independent of the network depth and the number of data points, and the bottleneck width enters only through scalar constraints. The construction proceeds by reducing the multilayer parameterization to a bilinear factorization, lifting it to a rank-constrained semidefinite program, expressing the rank constraint via a complementarity condition, and applying a completely positive lifting. While the resulting formulation is computationally intractable in general, it gives an exact conic representation of the nonconvexity induced by linear factorization and connects linear neural network training with copositive programming.

## 1 Introduction

Consider the training problem of an $N$-layer linear neural network:

$$\min_{W_1,\dots,W_N} \ \frac{1}{2n}\big\|W_N W_{N-1}\cdots W_1 X - Y\big\|_F^2, \tag{1}$$

where $X \in \mathbb{R}^{d_0 \times n}$ denotes the input data, $Y \in \mathbb{R}^{d_N \times n}$ the targets, and $W_k \in \mathbb{R}^{d_k \times d_{k-1}}$ the layer weight matrices. The network output is given by $\hat{Y} = W_N W_{N-1}\cdots W_1 X$.

Although the input-output map is linear, the multilayer parameterization induces a nonconvex optimization problem due to the factorized structure of the weights. Since compositions of linear mappings remain linear, depth does not increase expressivity, and problem (1) is equivalent, at the level of represented functions (and hence also in terms of global optima) to a low-rank matrix factorization with bilinear structure (Saxe et al., 2014; Kawaguchi, 2016).

In this work, we show that this nonconvex problem admits an exact convex reformulation. Specifically, we establish that it is equivalent to a convex optimization problem over a completely positive cone generated by positive semidefinite matrices in a lifted space. This formulation captures the nonconvexity induced by the factorization through a higher-dimensional conic representation. While the resulting formulation is convex, it involves a cone that is computationally intractable in general. The value of this reformulation is structural, as it provides an exact geometric characterization of the nonconvexity induced by factorization.

### 1.1 Related Work

Convex formulations of neural networks have been studied through infinite-dimensional function-space representations (Bengio et al., 2005; Bach, 2017), as well as through convex descriptions of overparameterized feature distributions (Fang et al., 2022). This perspective connects neural networks with reproducing kernel Banach spaces (Spek et al., 2025), data-fitting problems over infinite-dimensional spaces (Rosset et al., 2007;

Parhi & Nowak, 2021; Shenouda et al., 2024), and function approximation theory (Barron, 1993; Mhaskar, 2004; Meng & Ming, 2022; Liu et al., 2025).

Recent work has shown that certain neural network training problems admit exact finite-dimensional convex formulations. Convex reformulations for the nonconvex training of two-layer ReLU networks have been derived using convex semi-infinite duality and enumeration of ReLU sign patterns (Sahiner et al., 2021). Related techniques have also led to convex formulations for two- and three-layer convolutional neural networks with ReLU activations (Ergen & Pilanci, 2021a). For two-layer ReLU networks with weight decay, the training problem can be reformulated as a convex optimization problem with structured sparsity regularization (Pilanci & Ergen, 2020). Extensions to multiple three-layer ReLU regularized subnetworks have shown that such models can be equivalently cast as convex optimization problems in higher-dimensional spaces (Ergen & Pilanci, 2021c). Further results characterize optimal hidden-layer weights for two-layer and certain deep ReLU networks as extreme points of suitable convex sets (Ergen & Pilanci, 2021b;d). Subsequent work has studied the geometry of these formulations and the associated convex landscapes (Ergen & Pilanci, 2025). These ideas are also applied to attention and related token-mixing modules in vision transformers, yielding finite-dimensional convex formulations for self-attention, MLP-Mixer, and Fourier Neural Operator mechanisms (Sahiner et al., 2022).

Another major approach constructs convex relaxations through lifting techniques. These methods represent parameters through higher-order moment matrices or Gram matrices, leading to semidefinite or completely positive programs. Semidefinite liftings have been explored for training neural networks with polynomial activations (Bartan & Pilanci, 2025). This line of work was later extended to quantized neural networks with polynomial activations (Bartan & Pilanci, 2021), and to certain deep neural networks with polynomial and ReLU activations (Bartan & Pilanci, 2023). More recent work has explored convex formulations based on lifted representations for wide and shallow ReLU networks (Prakhya et al., 2025).

Compared with the approaches above, our construction avoids activation-pattern indexing, whose size depends on the data geometry, for example through $\mathrm{rank}(X)$. It gives an exact reformulation, and unlike lifted wide-network formulations, it applies to linear networks of arbitrary width, with the width appearing only through scalar constraints while the lifted cone dimension is determined by the input and output dimensions.

## 1.2 Notation

We denote by $\mathbb{S}^p$ the space of $p \times p$ real symmetric matrices, and by $\mathbb{S}^p_+$ its positive semidefinite cone. For symmetric matrices $A, B \in \mathbb{S}^p$, we write $A \succeq B$ (or $A \preceq B$) if $A - B$ is positive semidefinite. We write $\mathrm{vec}(A)$ for the vector obtained by stacking the columns of $A$, and $\mathrm{mat}(\cdot)$ for its inverse, i.e., for $x \in \mathbb{R}^{d^2}$, $\mathrm{mat}(x) \in \mathbb{R}^{d \times d}$ satisfies $\mathrm{vec}(\mathrm{mat}(x)) = x$. For inner product spaces $\mathbb{V}_1$ and $\mathbb{V}_2$, we write $\mathbb{V}_1 \oplus \mathbb{V}_2$ for their direct sum, equipped with the natural inner product. We use $\otimes$ to denote both the tensor and Kronecker product, depending on context.

We work with lifted variables of the form $z \otimes z$ with $z \in \mathbb{V}$, which lie in the symmetric subspace of $\mathbb{V} \otimes \mathbb{V}$, i.e., the subspace invariant under exchanging the two tensor factors $(u \otimes v) \mapsto (v \otimes u)$, denoted by $\mathrm{Sym}^2(\mathbb{V})$. In particular, for $\mathbb{V} = \mathbb{R}^p$, this subspace can be identified with $\mathbb{S}^p$, and the tensor $z \otimes z$ corresponds to the rank-one matrix $zz^\top$ under this identification.

## 2 Exact Convex Reformulation

Our main result establishes an exact convex reformulation for the training problem of a deep linear neural network in (1) based on generalized completely positive cones. We first recall the necessary definitions.

**Definition 1.** The *completely-positive cone* is defined as $\mathcal{CP}^p := \mathrm{conv}\{zz^\top : z \in \mathbb{R}^p_+\}$. Its dual cone, the *copositive cone*, is defined as $\mathcal{COP}^p = \{A \in \mathbb{S}^p : z^\top A z \geq 0, \ \forall z \in \mathbb{R}^p_+\}$.

Clearly, every completely positive matrix is positive semidefinite, and every positive semidefinite matrix is copositive, thus $\mathcal{CP}^p \subseteq \mathbb{S}^p_+ \subseteq \mathcal{COP}^p$.

This notion extends naturally to general cones (Bai et al., 2016; Nishijima & Nakata, 2024).

**Definition 2.** Let $\mathbb{K} \subset \mathbb{V}$ be a closed convex cone in the inner product space $\mathbb{V}$. The $\mathbb{K}$-*completely-positive cone* is defined as

$$\mathcal{CP}_{\mathbb{K}} := \operatorname{conv}\{z \otimes z : z \in \mathbb{K}\}.$$

Its dual, $\mathcal{COP}_{\mathbb{K}} = \{A \in \operatorname{Sym}^2(\mathbb{V}) : \langle A, z \otimes z \rangle \geq 0, \ \forall\, z \in \mathbb{K}\}$, is called the $\mathbb{K}$-*copositive cone.*

Completely positive cones arise naturally in the convex reformulations of quadratic optimization problems. They capture the convex hull of rank-one outer products over a cone, and thus provide a useful tool for lifted representation of certain nonconvex quadratic structures. We are now ready to introduce the lifted formulation.

**Theorem 1** (Exact CP reformulation). *Let $d := d_N + d_0$ and $r := \min_k d_k$. Define the selection matrices*

$$P_u = \begin{bmatrix} I_{d_N} \\ 0 \end{bmatrix} \in \mathbb{R}^{d \times d_N}, \quad P_v = \begin{bmatrix} 0 \\ I_{d_0} \end{bmatrix} \in \mathbb{R}^{d \times d_0}. \tag{2}$$

*Define $\widehat{\mathbb{K}} := \mathbb{R}_+ \times \operatorname{vec}(\mathbb{S}_+^d) \times \operatorname{vec}(\mathbb{S}_+^d) \times \operatorname{vec}(\mathbb{S}_+^d)$, where we used $\operatorname{vec}(\mathbb{S}_+^d)$ to denote the image of $\mathbb{S}_+^d$ under the $\operatorname{vec}(\cdot)$ operator. Then, problem* (1) *is equivalent to the following convex program:*

$$\min_{z_i, Z_{ij}} \quad \frac{1}{2n}\left( \langle P_v X X^\top P_v^\top \otimes P_u P_u^\top, Z_{11} \rangle - 2\langle \operatorname{vec}(P_u Y X^\top P_v^\top), z_1 \rangle + \|Y\|_F^2 \right)$$

$$\begin{aligned}
\text{s.t.} \quad & \operatorname{tr}(\operatorname{mat}(z_2)) = r, \\
& \langle \operatorname{vec}(I_d)\operatorname{vec}(I_d)^\top, Z_{22} \rangle = r^2, \\
& z_2 + z_3 = \operatorname{vec}(I_d), \\
& \operatorname{diag}(Z_{22} + Z_{23} + Z_{32} + Z_{33}) = \operatorname{vec}(I_d), \\
& \operatorname{tr}(\operatorname{mat}(z_1)) - \operatorname{tr}(Z_{12}) \leq 0, \\
& \begin{bmatrix} 1 & z_1^\top & z_2^\top & z_3^\top \\ z_1 & Z_{11} & Z_{12} & Z_{13} \\ z_2 & Z_{21} & Z_{22} & Z_{23} \\ z_3 & Z_{31} & Z_{32} & Z_{33} \end{bmatrix} \in \mathcal{CP}_{\widehat{\mathbb{K}}}
\end{aligned}$$

*in the sense that it has the same optimal value as* (1).

The proof proceeds in three steps. First, we reduce the deep linear network training problem to an equivalent shallow factorization, and we express the bilinear factorization through a rank-constrained positive semidefinite matrix. Second, we reformulate the rank constraint as a quadratic complementarity condition. Finally, we lift this formulation to a completely positive program over the cone $\mathcal{CP}_{\widehat{\mathbb{K}}}$.

## 2.1 Rank-Constrained Semidefinite Reformulation

Recall the model problem in (1) for deep linear neural networks. We first pass to the equivalent shallow factorization

$$\min_{\substack{U \in \mathbb{R}^{d_N \times r} \\ V \in \mathbb{R}^{d_0 \times r}}} \quad \frac{1}{2n} \|UV^\top X - Y\|_F^2, \tag{3}$$

where $r = \min_k d_k$. This equivalence is standard for deep linear networks (Saxe et al., 2014; Kawaguchi, 2016), and it follows from the fact that the composition $W_N \cdots W_1$ defines a linear map of rank at most $r$, while any matrix of rank at most $r$ admits a factorization of the form $UV^\top$.

This bilinear structure can be lifted to a rank-constrained positive semidefinite matrix. Define

$$W = \begin{bmatrix} U \\ V \end{bmatrix} \begin{bmatrix} U \\ V \end{bmatrix}^\top = \begin{bmatrix} UU^\top & UV^\top \\ VU^\top & VV^\top \end{bmatrix} \in \mathbb{S}_+^d,$$

so that we obtain $P_u^\top W P_v = UV^\top$ by using the block selection matrices $P_u$ and $P_v$ from (2). This leads to the following rank-constrained semidefinite programming formulation

$$\min_{W \in \mathbb{S}^d} \quad \frac{1}{2n} \|P_u^\top W P_v X - Y\|_F^2$$
$$\text{s.t.} \quad \text{rank}(W) \leq r, \quad W \succeq 0. \tag{4}$$

Problems in (3) and (4) are equivalent based on the following standard result.

**Lemma 1.** *Define the symmetric positive semidefinite block matrix $W = \begin{bmatrix} A & M \\ M^\top & B \end{bmatrix} \in \mathbb{S}_+^{d_N + d_0}$ with $A \in \mathbb{S}_+^{d_N}$ and $B \in \mathbb{S}_+^{d_0}$. Fix $r \in \mathbb{N}$, then the following sets are equivalent up to factorization via $M = UV^\top$:*

$$\left\{ (U, V) : U \in \mathbb{R}^{d_N \times r}, \ V \in \mathbb{R}^{d_0 \times r} \right\} \longleftrightarrow \left\{ W \in \mathbb{S}_+^{d_N + d_0} : \text{rank}(W) \leq r \right\}.$$

*Proof.* ($\Rightarrow$) Given $(U, V)$, set $Z := \begin{bmatrix} U \\ V \end{bmatrix} \in \mathbb{R}^{(d_N + d_0) \times r}$. Then $W = ZZ^\top \succeq 0$ with $\text{rank}(W) \leq r$, and its blocks satisfy $A = UU^\top$, $B = VV^\top$, $M = UV^\top$.

($\Leftarrow$) Given $W \succeq 0$ with $\text{rank}(W) \leq r$, take any factor $W = ZZ^\top$ with $Z \in \mathbb{R}^{(d_N + d_0) \times r}$ and partition $Z = \begin{bmatrix} U \\ V \end{bmatrix}$. Then $A = UU^\top$, $B = VV^\top$, and $M = UV^\top$. □

**Remark 1** (Expressivity vs. optimization bias)**.** Although both (1) and (3) represent the same family of linear mappings, the deep parameterization can induce distinct optimization dynamics and implicit regularization (Arora et al., 2019; Feng et al., 2022). Our focus here is on expressivity and on the equivalence of training problems at the level of global optima.

## 2.2 Complementarity Reformulation

We next reformulate the rank constraint via a complementarity condition.

**Lemma 2.** *A matrix $W \in \mathbb{S}_+^d$ satisfies $\text{rank}(W) \leq r$ for some $r \in \{0, \ldots, d\}$ if and only if there exists $W' \in \mathbb{S}_+^d$ such that*

$$\text{tr}(W') = r, \quad 0 \preceq W' \preceq I, \quad \langle W, I - W' \rangle \leq 0. \tag{5}$$

*Proof.* ($\Rightarrow$) If $\text{rank}(W) \leq r$, choose an $r$-dimensional subspace containing $\text{Ran}(W)$ and let $W'$ be the orthogonal projector onto this subspace. Then $W'$ satisfies the conditions.

($\Leftarrow$) Suppose the conditions in (5) hold. Since $W \succeq 0$ and $I - W' \succeq 0$, we have $\langle W, I - W' \rangle \geq 0$. Together with $\langle W, I - W' \rangle \leq 0$, this gives $\langle W, I - W' \rangle = 0$, and hence $W(I - W') = 0$. This shows that $\text{Ran}(W) \subseteq \ker(I - W')$, i.e., the range of $W$ is contained in the eigenspace of $W'$ corresponding to eigenvalue 1. Since $0 \preceq W' \preceq I$, all eigenvalues of $W'$ are in $[0, 1]$, hence the multiplicity of the eigenvalue 1 is at most $\text{tr}(W') = r$. Thus, $\text{rank}(W) = \dim(\text{Ran}(W)) \leq \dim(\ker(I - W')) \leq r$. □

Using Lemma 2, the rank-constrained formulation (4) can be written equivalently as

$$\min_{S, W, W' \in \mathbb{S}^d} \quad \frac{1}{2n} \|P_u^\top W P_v X - Y\|_F^2$$
$$\text{s.t.} \quad \text{tr}(W') = r,$$
$$W' + S = I_d,$$
$$\langle W, S \rangle \leq 0, \tag{6}$$
$$W \succeq 0, \quad W' \succeq 0, \quad S \succeq 0.$$

In particular, (4) and (6) have the same feasible set in the variable $W$, and therefore the same optimal value.

## 2.3 Completely Positive Lifting

We now lift the complementarity formulation to a completely positive program. We first express the problem in vectorized form in order to apply Lemma 3. We then rewrite the resulting formulation in a matrix-based representation using Kronecker products, which provides a more natural interpretation of the lifted variables.

### 2.3.1 Vectorized Formulation

Problem (6) admits an exact reformulation as a completely positive program via a quadratic lifting. The following lemma is a simplified specialization of Theorem 4 in (Bai et al., 2016), stated in the form needed for our application.

**Lemma 3** (Completely positive lifting). *Let $\mathbb{K} \subset \mathbb{R}^p$ be a closed convex cone and define the augmented cone $\widehat{\mathbb{K}} := \mathbb{R}_+ \times \mathbb{K}$. Let $h, \bar{h} \in \mathbb{R}$, $q, \bar{q} \in \mathbb{R}^p$, and $Q, \bar{Q} \in \mathbb{R}^{p \times p}$ be given, with $Q \in \mathcal{COP}_{\mathbb{K}}$. Let $A \in \mathbb{R}^{m \times p}$ and $b \in \mathbb{R}^m$, and write $a_i^\top$ for the i-th row of A and $b_i$ for the corresponding entry of b. Consider the quadratically constrained quadratic program*

$$
\begin{aligned}
\min_{z \in \mathbb{R}^p} \quad & z^\top Q z + q^\top z + h \\
\text{s.t.} \quad & a_i^\top z = b_i, \quad for\ i = 1 \ldots, m \\
& z^\top \bar{Q} z + \bar{q}^\top z + \bar{h} \leq 0, \\
& z \in \mathbb{K},
\end{aligned}
\tag{7}
$$

*and suppose that its feasible set contains no point at which the quadratic constraint is satisfied strictly. Then, (7) is equivalent to the following completely positive program:*

$$
\begin{aligned}
\min_{z \in \mathbb{R}^p, Z \in \mathbb{S}^p} \quad & \langle Q, Z \rangle + q^\top z + h \\
\text{s.t.} \quad & a_i^\top z = b_i, \quad for\ i = 1, \ldots, m \\
& \langle a_i a_i^\top, Z \rangle = b_i^2, \quad for\ i = 1, \ldots, m \\
& \langle \bar{Q}, Z \rangle + \bar{q}^\top z + \bar{h} \leq 0, \\
& \begin{bmatrix} 1 & z^\top \\ z & Z \end{bmatrix} \in \mathcal{CP}_{\widehat{\mathbb{K}}},
\end{aligned}
\tag{8}
$$

*in the sense that problems (7) and (8) have the same optimal value. Moreover, if $(z, Z)$ is optimal for (8), then z belongs to the convex hull of the set of optimal solutions of (7).*

We now apply Lemma 3 to the complementarity reformulation in (6). To this end, we identify the finite-dimensional matrix space $\mathbb{S}^d \times \mathbb{S}^d \times \mathbb{S}^d$ with a Euclidean space via vectorization, so that the complementarity formulation fits the setting of Lemma 3. Define

$$
z := \begin{bmatrix} \mathrm{vec}(W) \\ \mathrm{vec}(W') \\ \mathrm{vec}(S) \end{bmatrix} \in \mathbb{R}^{3d^2}, \qquad \mathbb{K} := \mathrm{vec}(\mathbb{S}_+^d) \times \mathrm{vec}(\mathbb{S}_+^d) \times \mathrm{vec}(\mathbb{S}_+^d) \subset \mathbb{R}^{3d^2},
$$

where $\mathrm{vec}(\mathbb{S}_+^d)$ denotes the image of $\mathbb{S}_+^d$ under the $\mathrm{vec}(\cdot)$ operator. We introduce new block-selection matrices

$$
\begin{aligned}
R_1 &:= \begin{bmatrix} I_{d^2} & 0_{d^2} & 0_{d^2} \end{bmatrix} \in \mathbb{R}^{d^2 \times 3d^2} \\
R_2 &:= \begin{bmatrix} 0_{d^2} & I_{d^2} & 0_{d^2} \end{bmatrix} \in \mathbb{R}^{d^2 \times 3d^2} \\
R_3 &:= \begin{bmatrix} 0_{d^2} & 0_{d^2} & I_{d^2} \end{bmatrix} \in \mathbb{R}^{d^2 \times 3d^2}
\end{aligned}
$$

so that $R_1 z = \mathrm{vec}(W)$, $R_2 z = \mathrm{vec}(W')$ and $R_3 z = \mathrm{vec}(S)$.

We are now ready to express problem (6) in terms of $z$.

**Proposition 1.** *Problem (6) is equivalent to problem (7) with*

$$
Q = \frac{1}{2n} R_1^\top \big( (P_v X X^\top P_v^\top) \otimes (P_u P_u^\top) \big) R_1, \qquad \bar{Q} = -\frac{1}{2}\big( R_1^\top R_2 + R_2^\top R_1 \big), \qquad A = \begin{bmatrix} \mathrm{vec}(I_d)^\top R_2 \\ R_2 + R_3 \end{bmatrix}
$$

$$
q = -\frac{1}{n} R_1^\top \mathrm{vec}(P_u Y X^\top P_v^\top), \qquad\qquad \bar{q} = R_1^\top \mathrm{vec}(I_d), \qquad\qquad b = \begin{bmatrix} r \\ \mathrm{vec}(I_d) \end{bmatrix}.
$$

$$
h = \frac{1}{2n} \|Y\|_F^2, \qquad\qquad\qquad \bar{h} = 0,
$$

*Moreover, $Q \in \mathcal{COP}_{\mathbb{K}}$ and the quadratic constraint satisfies the non-strict feasibility condition in Lemma 3. Therefore, the assumptions are satisfied, and the formulation is equivalent to the lifted problem* (8).

*Proof.* Using the identity $\text{vec}(AWB) = (B^\top \otimes A)\text{vec}(W)$, we have $\text{vec}(P_u^\top W P_v X) = (X^\top P_v^\top \otimes P_u^\top)R_1 z$.

Expanding the objective, we obtain

$$\frac{1}{2n}\|P_u^\top W P_v X - Y\|_F^2 = \frac{1}{2n}\left(\|P_u^\top W P_v X\|_F^2 - 2\langle P_u^\top W P_v X, Y\rangle + \|Y\|_F^2\right).$$

The quadratic term satisfies

$$\begin{aligned}
\|P_u^\top W P_v X\|_F^2 &= \text{vec}(P_u^\top W P_v X)^\top \text{vec}(P_u^\top W P_v X) \\
&= z^\top R_1^\top (X^\top P_v^\top \otimes P_u^\top)^\top (X^\top P_v^\top \otimes P_u^\top) R_1 z \\
&= z^\top R_1^\top (P_v X \otimes P_u)(X^\top P_v^\top \otimes P_u^\top) R_1 z \\
&= z^\top R_1^\top (P_v X X^\top P_v^\top \otimes P_u P_u^\top) R_1 z.
\end{aligned}$$

For the linear term, we have

$$\langle P_u^\top W P_v X, Y\rangle = \langle W, P_u Y X^\top P_v^\top\rangle = z^\top R_1^\top \text{vec}(P_u Y X^\top P_v^\top).$$

Hence we obtain the terms $Q$, $q$ and $h$ in Proposition 1. Moreover, since $(P_v X X^\top P_v^\top) \succeq 0$ and $(P_u P_u^\top) \succeq 0$, their Kronecker product is also positive semidefinite, hence $Q \succeq 0$ and therefore $Q \in \mathcal{COP}_{\mathbb{K}}$.

Looking at the affine constraints, we have $\text{tr}(W') = \langle I_d, W'\rangle = \text{vec}(I_d)^\top R_2 z$ and $S + W' = I_d$. Hence, it can be written in the form $Az = b$, with $A$ and $b$ as defined in Proposition 1.

Finally, we reformulate the quadratic constraint by

$$\begin{aligned}
\langle W, I_d - W'\rangle &= \langle W, I_d\rangle - \frac{1}{2}\left(\langle W, W'\rangle + \langle W', W\rangle\right) \\
&= \text{vec}(I_d)^\top R_1 z - \frac{1}{2}\left(z^\top R_1^\top R_2 z + z^\top R_2^\top R_1 z\right)
\end{aligned}$$

leading to the terms $\bar{Q}$, $\bar{q}$, $\bar{h}$ in Proposition 1. The quadratic constraint is satisfied at equality for all feasible points of (6), hence the non-strict feasibility condition holds. This establishes the equivalence. $\square$

The problem formulation in Theorem 1 is an algebraic reformulation of Proposition 1, expressed in a more structured and interpretable form.

### 2.3.2 Kronecker Representation

The same construction admits a tensor interpretation. In this subsection, we use this identification to write the lifted blocks in Kronecker form.

Let $\bar{\mathbb{K}} = \mathbb{R}_+ \oplus \mathbb{S}_+^d \oplus \mathbb{S}_+^d \oplus \mathbb{S}_+^d$. Consider $\mathcal{W} \in \mathcal{CP}_{\bar{\mathbb{K}}}$ with the blocks corresponding to the components $(\omega_0, \omega_1, \omega_2, \omega_3) \in \bar{\mathbb{K}}$. The vectorization map identifies $\bar{\mathbb{K}}$ with the cone $\widehat{\mathbb{K}}$ used in Theorem 1. Under this identification, a vectorized lifted block $Z_{ij}$ is equivalently a matrix-indexed block $\mathcal{W}_{ij}$. On rank-one atoms, with $\omega_0 \equiv 1$, $\omega_1 \equiv W$, $\omega_2 \equiv W'$, and $\omega_3 \equiv S$, the corresponding lifted blocks are $\mathcal{W}_{ij} = \omega_i \otimes \omega_j$, e.g.,

$$\mathcal{W}_{01} = W, \quad \mathcal{W}_{02} = W', \quad \mathcal{W}_{03} = S, \quad \mathcal{W}_{11} = W \otimes W, \quad \mathcal{W}_{12} = W \otimes W', \quad \mathcal{W}_{22} = W' \otimes W', \quad \mathcal{W}_{23} = W' \otimes S.$$

General elements of the cone are convex combinations of such atoms.

To express the vectorized quadratic coefficient against standard Kronecker-product lifted blocks, we use the following reshuffling operator. For $M \in \mathbb{R}^{d^2 \times d^2}$, define

$$\left(\mathcal{R}(M)\right)_{(i,k),(j,\ell)} = M_{(i,j),(k,\ell)}, \quad \text{for all } i, j, k, \ell.$$

$\mathcal{R}$ is a permutation of the entries of its matrix argument, i.e., there exists a permutation matrix $\Pi \in \mathbb{R}^{d^4 \times d^4}$ such that $\text{vec}(\mathcal{R}(M)) = \Pi \, \text{vec}(M)$. It changes the matricization of a fourth-order tensor from the $(i,j),(k,\ell)$ grouping induced by $\text{vec}(W)\,\text{vec}(W)^\top$ to the $(i,k),(j,\ell)$ grouping induced by the standard Kronecker product $W \otimes W$. In particular,

$$\langle \mathcal{R}(B \otimes A), W \otimes W \rangle = \langle B \otimes A, \text{vec}(W)\,\text{vec}(W)^\top \rangle.$$

We are now ready to present our Kronecker formulation.

**Theorem 2** (Exact CP reformulation). *Problem* (1) *is equivalent to the following convex program:*

$$
\begin{aligned}
\min_{\mathcal{W}_{ij}} \quad & \frac{1}{2n}\left( \langle \mathcal{R}(P_v X X^\top P_v^\top \otimes P_u P_u^\top), \mathcal{W}_{11}\rangle - 2\langle P_u Y X^\top P_v^\top, \mathcal{W}_{01}\rangle + \|Y\|_F^2 \right) \\
\text{s.t.} \quad & \text{tr}(\mathcal{W}_{02}) = r, \\
& \text{tr}(\mathcal{W}_{22}) = r^2, \\
& \langle E_{ij}, \mathcal{W}_{02} + \mathcal{W}_{03}\rangle = \delta_{ij}, \quad \forall i,j, \\
& \langle E_{ij} \otimes E_{ij}, \mathcal{W}_{22} + \mathcal{W}_{23} + \mathcal{W}_{32} + \mathcal{W}_{33}\rangle = \delta_{ij}, \quad \forall i,j, \\
& \text{tr}(\mathcal{W}_{01}) - \langle \Delta, \mathcal{W}_{12}\rangle \leq 0, \\
& \begin{pmatrix} 1 & \mathcal{W}_{01} & \mathcal{W}_{02} & \mathcal{W}_{03} \\ \mathcal{W}_{10} & \mathcal{W}_{11} & \mathcal{W}_{12} & \mathcal{W}_{13} \\ \mathcal{W}_{20} & \mathcal{W}_{21} & \mathcal{W}_{22} & \mathcal{W}_{23} \\ \mathcal{W}_{30} & \mathcal{W}_{31} & \mathcal{W}_{32} & \mathcal{W}_{33} \end{pmatrix} \in \mathcal{CP}_{\bar{\mathbb{K}}},
\end{aligned}
\tag{9}
$$

*in the sense that* (9) *has the same optimal value as* (1). *Here* $\delta_{ij}$ *is the indicator that takes* 1 *if* $i = j$ *and* 0 *otherwise,* $\{E_{ij}\}_{i,j}$ *denotes the standard matrix basis,* $E_{ij} \otimes E_{ij}$ *denotes the corresponding Kronecker basis elements, and* $\Delta = \sum_{i=1}^d \sum_{j=1}^d E_{ij} \otimes E_{ij}$ *is the diagonal contraction operator satisfying* $\langle \Delta, A \otimes B \rangle = \langle A, B \rangle$ *for all* $A, B \in \mathbb{R}^{d \times d}$. *The formulation is linear in the lifted variable* $\mathcal{W}$, *with all nonconvexity encoded in the completely positive constraint.*

*Proof.* We prove the equivalence term by term. Since both the formulations are linear in the lifted variables, it is enough to verify the identities on rank-one atoms and then extend by convexity. Consider therefore an atom generated by $\omega = (1, W, W', S)$.

First consider the objective. For the quadratic term,

$$
\begin{aligned}
\langle P_v X X^\top P_v^\top \otimes P_u P_u^\top, Z_{11}\rangle &= \langle P_v X X^\top P_v^\top \otimes P_u P_u^\top, \text{vec}(W)\,\text{vec}(W)^\top \rangle \\
&= \langle \mathcal{R}(P_v X X^\top P_v^\top \otimes P_u P_u^\top), W \otimes W \rangle \\
&= \langle \mathcal{R}(P_v X X^\top P_v^\top \otimes P_u P_u^\top), \mathcal{W}_{11} \rangle.
\end{aligned}
$$

The linear term is unchanged by the matrix-indexed notation because the first-order block satisfies $\mathcal{W}_{01} = W$ on atoms:

$$\langle P_u^\top W P_v X, Y \rangle = \langle W, P_u Y X^\top P_v^\top \rangle = \langle P_u Y X^\top P_v^\top, \mathcal{W}_{01}\rangle.$$

This proves the equivalence of the objective terms.

Next consider the affine constraints. The trace constraint on $W'$ becomes $\text{tr}(W') = \text{tr}(\mathcal{W}_{02}) = r$.

The equality $W' + S = I_d$ is equivalent to the componentwise equations $\langle E_{ij}, W' + S \rangle = \delta_{ij}$, for $i, j = 1, \ldots, d$, which takes the form $\langle E_{ij}, \mathcal{W}_{02} + \mathcal{W}_{03}\rangle = \delta_{ij}$.

The CP lifting of these affine equations adds the squared constraints. For $\text{tr}(W') = r$, the lifted condition is

$$
\begin{aligned}
\langle \text{vec}(I_d)\,\text{vec}(I_d)^\top, Z_{22}\rangle &= \langle \text{vec}(I_d)\,\text{vec}(I_d)^\top, \text{vec}(W')\,\text{vec}(W')^\top \rangle \\
&= \langle I_d, W' \rangle \langle I_d, W' \rangle \\
&= \langle I_d \otimes I_d, W' \otimes W' \rangle \\
&= \langle I_d \otimes I_d, \mathcal{W}_{22}\rangle = \text{tr}(\mathcal{W}_{22}) = r^2.
\end{aligned}
$$

Similarly, the affine equation $\langle E_{ij}, W' + S \rangle = \delta_{ij}$ has lifted square $\left(\langle E_{ij}, W' + S \rangle\right)^2 = \delta_{ij}^2 = \delta_{ij}$. Expanding the square in lifted variables gives

$$\langle E_{ij} \otimes E_{ij}, W' \otimes W' + W' \otimes S + S \otimes W' + S \otimes S \rangle = \langle E_{ij} \otimes E_{ij}, \mathcal{W}_{22} + \mathcal{W}_{23} + \mathcal{W}_{32} + \mathcal{W}_{33} \rangle = \delta_{ij}.$$

It remains to identify the complementarity inequality. By definition of $\Delta$, we have on rank-one atoms that

$$\text{tr}(\mathcal{W}_{01}) - \langle \Delta, \mathcal{W}_{12} \rangle = \langle I_d, W \rangle - \langle W, W' \rangle = \langle W, I_d - W' \rangle.$$

Using the affine constraint $S = I_d - W'$, this is precisely $\langle W, S \rangle \leq 0$, the complementarity inequality in (6).

All identities above agree with the corresponding vectorized constraints on every rank-one generating atom. Since both formulations impose the same linear constraints on convex combinations of these atoms and use the same completely positive cone under the vectorization isomorphism, their feasible sets correspond bijectively under reshaping. The objective values are preserved by the same identities, so the two formulations are equivalent. $\qquad\square$

## 3 Conclusion

We have shown that the squared-loss training problem for deep linear neural networks admits an exact convex reformulation after lifting to a generalized completely positive cone. The resulting convex program has the same global optimal value as the original nonconvex training problem, and the size of the lifted representation depends on the input and output dimensions rather than the network depth or number of samples, with the bottleneck width appearing only in scalar constraints.

The reformulation is not intended as a scalable algorithm, since membership in the generalized completely positive cone is intractable in general. Its main value is instead structural, as it identifies the precise conic object capturing the nonconvexity induced by linear factorization. This connection suggests two natural directions for future work, which are developing tractable outer approximations that preserve useful information about the training objective and understanding whether analogous liftings can provide meaningful convex descriptions for broader neural network architectures.

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
