# OpenReview forum: "Exact Convex Reformulations of Linear Neural Networks via Completely Positive Lifting"
_TMLR — Under review for TMLR_

### Review · Reviewer_YsWk · 2026-07-01

**Summary Of Contributions:**

[summary]

The submission studies squared-loss training of deep linear neural networks. Since the product of linear layers is a single linear map with rank at most the bottleneck width, the authors reduce the training problem to a shallow bilinear factorization of $U V^T$. They then lift the bilinear formulation to a rank-constrained PSD program, rewrite the rank constraint using a PSD complementarity condition, and finally apply a generalized completely positive lifting to obtain an exact convex conic reformulation. The final result has the same optimal value as the original nonconvex training problem, with the nonconvexity encoded in a generalized CP cone.

The dimension observation in the final theorem states an interesting observation that the data enters only through $XX^T$ and $YX^T$, while the lifted cone dimension depends on $d_0+d_N$, not on the number of samples or network depth.


[strength]

1.	The proof strategy is easy to follow: deep linear network -> bilinear low-rank factorization -> rank-constrained SDP -> complementarity -> completely positive lift.

2.	The core rank-complementarity lemma is simple and useful in the sense that it offers a clean bridge from the rank constraint to a conic QP with complementarity.

3.	The dimension observation in the final theorem is appealing in that the data enter only through $XX^T$ and $YX^T$, while the lifted cone dimension depends on $d_0+d_N$, not on the number of samples or network depth. The conclusion states this clearly.

4.	The authors are honest about tractability. The abstract and conclusion both say the CP formulation is computationally intractable in general and should be viewed as structural rather than algorithmic.

[weakness]

1.	The main proof relies on a known CP lifting result for conic QCQPs, stated as a specialization of Bai et al. [1] cited in the submission. Since deep linear training under squared loss reduces to a low-rank regression / rank-constrained least-squares problem, the paper needs a much sharper explanation of what is new beyond applying a generic completely positive reformulation to a known QCQP. Even though TMLR does not look for high impact and significance, I believe the manuscript should still distinguish itself from rebranding of existing works.

2.	The “no strictly feasible point” condition in Lemma  3 (which turn QCQP into CP) is unusual. The paper should either reproduce the proof in this specialized setting or quote the exact theorem from Bai et al. [1] with all assumptions. Otherwise, we have to trust that no closure, attainment, homogenization, or recession-cone issue is hidden by the simplified statement.

3.	The phrase “exact convex reformulation” is potentially misleading. The formulation preserves the optimal value, but because the lifted optimum may correspond to a convex combination of original optimizers, it does not necessarily recover a single feasible low-rank matrix or a set of network weights without additional decomposition

4.	I feel that Theorem 1 and Theorem 2 are reshaped version of each other. This adds notation without much new content. The paper would be stronger if it kept one main theorem and moved the alternate representation to a remark.

[1] Bai, L., Mitchell, J. E., & Pang, J. S. (2016). On conic QPCCs, conic QCQPs and completely positive programs. Mathematical Programming, 159(1), 109-136.

**Additional Comments:**

The paper should set its distinction from existing work on CP lifting for conic QCQPs.

**Audience:**

Yes

**Audience Explanation:**

Readers working on optimization theory, convex reformulations, and theoretical properties of neural network training would find this paper relevant and interesting.

**Broader Impact Concerns:**

I do not see a concern that would necessitate a dedicated broader impact statement. The paper is a theoretical work and does not introduce any dataset, human-subject study, or capability with obvious direct societal or ethical risk.

**Claims And Evidence:**

Yes

**Claims Explanation:**

The mathematical claims appear accurate and reasonably well supported.
The paper’s main claim is an exact convex reformulation of deep linear network training via completely positive lifting, and the proof structure is coherent: it reduces the deep linear network to a rank-constrained shallow factorization, rewrites this as a rank-constrained semidefinite program, expresses the rank constraint through a complementarity condition, and then applies completely positive lifting.

**Requested Changes:**

[question]

1.	Why the author choose full vectorization rather than a symmetric-vectorization convention, and they should carefully define all induced inner products. Full vectorization may duplicate off-diagonal degrees of freedom.

2.	Since the original problem has a closed-form global solution via reduced-rank regression / truncated SVD, what new information does the CP formulation provide?

3.	Can one recover an actual optimal set of network weights from a solution of the CP program?

[suggestions]

1.	It would be good if the author could develop relaxations, bounds, approximation hierarchies, dual certificates, or new theoretical consequences for linear-network training, following their acknowledgement of the intractability of the CP cone.

2.	The authors can provide one small worked example showing the lifted variables and constraints.

---

### Review · Reviewer_S5NH · 2026-07-04

**Summary Of Contributions:**

## Main Summmary

This paper shows that training a deep linear network with squared loss can be written exactly as a convex optimization problem after lifting to a generalized completely positive cone. The derivation goes through a few standard steps: first reducing the deep linear network to a low-rank bilinear factorization, then rewriting the rank constraint using a PSD matrix and a complementarity condition, and finally applying a completely positive lifting result.

The final formulation is convex in the lifted variables, although the cone itself is intractable in general. So I view the paper mainly as a structural result rather than an algorithmic contribution.

## Overall assessment

I think the main result is basically correct, and the proof strategy is clear. The paper gives a neat conic representation of the nonconvexity coming from linear factorization. That said, I also think the current version somewhat oversells the novelty unless the authors more carefully explain how much of the result follows from existing CP lifting machinery and how much is specific to deep linear networks.

## What I liked

The paper is easy to follow at a high level. The reduction from a deep linear network to a shallow low-rank factorization is standard, and the subsequent rank-constrained SDP formulation is natural. I also found the complementarity reformulation of the rank constraint clean. Once this is written as a conic QCQP/QPCC, the CP lifting step is fairly straightforward, but the paper still does a useful job putting these pieces together.

I also appreciate that the authors are explicit that the final convex formulation is not computationally practical. This avoids giving the impression that the paper provides a scalable training algorithm. The contribution is better understood as an exact structural representation of the training problem.

## Main concerns

My biggest concern is about novelty and positioning. Since deep linear network training with squared loss reduces to a low-rank least-squares or reduced-rank regression problem, the paper should discuss that literature more directly. Right now the related work focuses mostly on convex neural network reformulations, but the connection to classical reduced-rank regression and rank-constrained matrix optimization is underdeveloped.

Relatedly, the CP lifting part seems to rely heavily on existing results for conic QCQPs/QPCCs. That is fine, but the paper should be more explicit about what is new here. Is the main novelty the observation that deep linear network training can be put into exactly the right form for this lifting? Is it the particular way the bottleneck rank enters only through scalar constraints? Is it the depth-independent lifted representation? I think the paper would be stronger if it stated this more directly.

A second concern is the use of the word “equivalent.” The theorems show that the lifted convex problem has the same optimal value as the original nonconvex problem. This is an important statement, but it is weaker than saying the optimization problems are equivalent in a fully constructive sense. In particular, an optimal lifted solution may be a convex combination of rank-one atoms. The authors should clarify whether one can recover an optimal original network from an optimal lifted solution, and if so whether this recovery is constructive or only follows abstractly from a CP decomposition. If the paper only wants to claim equality of optimal values, the wording should be made consistent throughout.

A third issue is the dimension of the lifted formulation. The paper says the lifted dimension depends only on the input and output dimensions, and not on the depth or number of samples. This is true in the intended sense, but the actual lifted cone is still extremely large. Since the result is not computational, I think the paper should give a precise dimension and constraint count. This would help readers understand exactly what kind of structural reformulation is being claimed.

## Some smalller technical and presentation comments

The definition of the bottleneck rank (r) should be made completely explicit. The paper writes $r := \min_k d_k$. For the rank of $W_N \cdots W_1$, this should include the input and output dimensions as well, i.e. something like $\min_{0 \leq k \leq N} d_k$. If that is what the authors mean, they should say so clearly.

The Kronecker version in Theorem 2 is useful, but the notation is a bit dense. I would like the authors to spell out what space each block $W_{ij}$ belongs to, how the trace of blocks such as $W_{22}$ should be interpreted, and how the vectorized and matrix-indexed versions are related. The use of the full standard basis $E_{ij}$ for symmetric matrix variables may also create some notational redundancy or confusion, so it would be helpful to clarify this.

I would also suggest adding a very small example, perhaps with $d_0=d_N=2$, just to show what the variables and constraints look like. Since the paper is abstract and conic-optimization-heavy, a toy example would make the construction much easier to digest.

**Additional Comments:**

N/A

**Audience:**

Yes

**Audience Explanation:**

Readers interested in optimization theory, convex analysis, exact neural network reformulations, and the geometry of nonconvex training problems. For that audience, an exact CP/copo representation of deep linear network training is potentially interesting as a structural result.

**Broader Impact Concerns:**

I do not see any significant broader impact concerns. This is a theoretical paper about an exact conic reformulation of a simplified neural network training problem, and it does not introduce a new deployed system, dataset, or application.

**Claims And Evidence:**

Yes

**Claims Explanation:**

Yes, mostly. The main claim: deep linear network training under squared loss has the same optimal value as a convex problem over a generalized completely positive cone is supported by a reasonable proof chain: reduction to low-rank bilinear factorization, rank-constrained PSD formulation, complementarity reformulation, and CP lifting.

However, the paper should be more precise about the word “equivalent.” The current proof mainly establishes equality of optimal values, while recovery of an original network from a lifted solution is not fully explained. I would also like to see a clearer dimension count for the lifted cone and a bit more clarification of the Kronecker notation.

Overall, I find the main claim convincing, but the presentation should be tightened.

**Requested Changes:**

The main thing I would like the authors to clarify is the novelty of the result. Since deep linear network training with squared loss reduces to a low-rank least-squares or reduced-rank regression problem, the paper should discuss this classical connection more directly. Right now, the related work focuses more on convex neural network reformulations, but says less about older work on low-rank regression and matrix factorization.

I would also like the authors to be more precise about what “equivalent” means. The current result seems to show equality of optimal values. If an optimal lifted solution can be used to recover an optimal original network, the paper should explain how. If the recovery is only nonconstructive through a CP decomposition, that should also be stated clearly.

A few technical details should also be clarified. The definition of the bottleneck rank $r$ should explicitly say whether the minimum is over all layers, including input and output dimensions. The paper should also give a clear dimension and constraint count for the lifted formulation, since the CP cone is intractable and the size of the lift matters for understanding the result. Finally, the authors should explain more clearly which parts of the reformulation come from general CP lifting results and which parts are specific to deep linear networks.

Some smaller improvements would also help. A small worked example, for instance with $d_0=d_N=2$, would make the construction easier to follow. The Kronecker formulation could use clearer notation, especially for the spaces of the lifted blocks and the inner products. I would also suggest toning down phrases like “exact convex reformulation” in the introduction by immediately noting that the convexity is over an intractable CP cone. A short discussion of possible tractable relaxations or approximations would also make the future-work section more concrete.

---

### Review · Reviewer_Vd4b · 2026-07-12

**Summary Of Contributions:**

This paper studies the training of deep linear neural networks under squared loss and derives an exact convex reformulation over a generalized completely positive (CP) cone. The authors first reduce the multilayer parameterization to an equivalent low-rank bilinear factorization, which can be represented by a rank-constrained positive semidefinite programming (SDP). Then, they encode the rank constraint using a complementarity condition and apply a completely positive lifting. The resulting conic program is linear in the lifted variables and has the same global optimal value as the original training problem. Its ambient dimension depends only on the input and output dimensions, rather than on the network depth or number of training samples. The paper also presents an equivalent Kronecker-product representation of the lift.

**Strengths:** The reformulation provides a structural characterization of the nonconvexity induced by deep linear factorization, and it preserves the global value. Moreover, the bottleneck width appears only in scalar constraints. This finding could be helpful for future research in understanding deep linear neural network training.

**Weaknesses:** The generalized completely positive cone is computationally intractable, so it may have limited algorithmic value. Moreover, the underlying training problem is already tractable. Unregularized deep linear-network training under squared loss reduces to reduced-rank regression, whose global predictor can be obtained via standard singular-value-decomposition techniques. This substantially limits the computational significance of the convex reformulation. The conceptual novelty is somewhat incremental. The results rely on several technical tools from prior work, including the reduction to a low-rank linear map, Gram-matrix lifting, complementarity representation of rank, and exact completely positive reformulation of rank-constrained conic problems.

**Audience:**

Yes

**Audience Explanation:**

I think researchers working on the optimization foundations of machine learning, convex formulations of neural networks, and low-rank matrix factorization would likely be interested in this result. However, given its weaknesses, I think the impact of this work might be limited.

**Claims And Evidence:**

Yes

**Claims Explanation:**

I checked the proofs, and they are mostly mathematically sound. However, Theorem 2 has a gap: it treats only completely positive atoms with scalar coordinate 1, whereas the cone also contains zero-leading atoms representing recession directions. Note that the cone is
$conv(\omega\otimes\omega:\omega\in \mathbb R_+ \oplus\mathbb S_+^d\oplus\mathbb S_+^d\oplus\mathbb S_+^d)$, whose
generating atoms have the general form $\omega=(t,A,B,C), t\ge 0$ not only $t=1$.

**Requested Changes:**

* The authors should fix the gap in the proof of Theorem 2.

* The authors should more explicitly explain how Lemma 3 follows from Bai et al. The cited theorem is stated in different notation and under different hypotheses, including a condition on the recession cone and an equality form of the lifted quadratic constraint. The manuscript should provide a precise specialization showing that $Q\in COP_K$ implies the required copositivity condition on the relevant recession directions, and explain why the lifted inequality is forced to hold at equality under the no-strict-feasibility assumption.

* Bai et al. already provide exact completely positive formulations for broad classes that include rank-constrained semidefinite programs and conic complementarity problems. The novelty relative to general completely positive reformulations and the paper's contribution should be clarified more explicitly.

* Theorem 1 contains the main mathematical contribution. Theorem 2 is essentially a coordinate reshaping of that result. The exposition would benefit from explicitly stating this and from avoiding presenting the Kronecker formulation as a separate convexification principle.